# Expected Information Maximization Using the I-Projection for Mixture Density Estimation

**Philipp Becker**
Autonomous Learning Robots, KIT
Bosch Center for Artificial Intelligence
*Correspondence to: philipp.becker@kit.edu*

**Oleg Arenz**
Intelligent Autonomous Systems, TU Darmstadt

**Gerhard Neumann**
Autonomous Learning Robots, KIT
Bosch Center for Artificial Intelligence
University of Tübingen

## Abstract

Modelling highly multi-modal data is a challenging problem in machine learning. Most algorithms are based on maximizing the likelihood, which corresponds to the M(oment)-projection of the data distribution to the model distribution. The M-projection forces the model to average over modes it cannot represent. In contrast, the I(nformation)-projection ignores such modes in the data and concentrates on the modes the model can represent. Such behavior is appealing whenever we deal with highly multi-modal data where modelling single modes correctly is more important than covering all the modes. Despite this advantage, the I-projection is rarely used in practice due to the lack of algorithms that can efficiently optimize it based on data. In this work, we present a new algorithm called Expected Information Maximization (EIM) for computing the I-projection solely based on samples for general latent variable models, where we focus on Gaussian mixtures models and Gaussian mixtures of experts. Our approach applies a variational upper bound to the I-projection objective which decomposes the original objective into single objectives for each mixture component as well as for the coefficients, allowing an efficient optimization. Similar to GANs, our approach employs discriminators but uses a more stable optimization procedure, using a tight upper bound. We show that our algorithm is much more effective in computing the I-projection than recent GAN approaches and we illustrate the effectiveness of our approach for modelling multi-modal behavior on two pedestrian and traffic prediction datasets.

## 1 Introduction

Learning the density of highly multi-modal distributions is a challenging machine learning problem relevant to many fields such as modelling human behavior (Pentland & Liu, 1999). Most common methods rely on maximizing the likelihood of the data. It is well known that the maximum likelihood solution corresponds to computing the M(oment)-projection of the data distribution to the parametric model distribution (Bishop, 2006). Yet, the M-projection averages over multiple modes in case the model distribution is not rich enough to fully represent the data (Bishop, 2006). This averaging effect can result in poor models, that put most of the probability mass in areas that are not covered by the data. The counterpart of the M-projection is the I(nformation)-projection. The I-projection concentrates on the modes the model is able to represent and ignores the remaining ones. Hence, it does not suffer from the averaging effect (Bishop, 2006).

In this paper, we explore the I-projection for mixture models which are typically trained by maximizing the likelihood via expectation maximization (EM) (Dempster et al., 1977). Despite the richness of mixture models, the averaging problem remains as we typically do not know the correct number

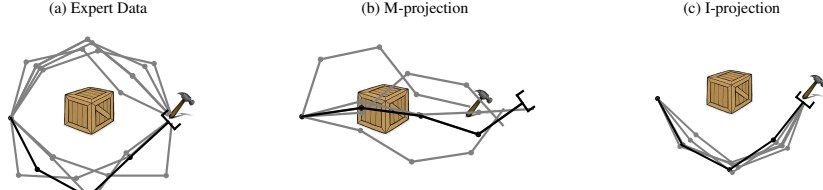

Figure 1: Illustration of the I-projection vs. the M-projection for modelling behavior. (a): A robot reaches a target point while avoiding an obstacle. There are two different types of solutions, above and below the obstacle. A single Gaussian is fitted to the expert data in joint space. (b): The M-projection fails to reach the target and collides with the obstacle. (c): The I-projection ignores the second mode and reaches the target while avoiding the obstacle.

of modes and it is hard to identify all modes of the data correctly. By the use of the I-projection, our mixture models do not suffer from this averaging effect and can generate more realistic samples that are less distinguishable from the data. In this paper we concentrate on learning Gaussian mixture models and conditional Gaussian mixtures of experts (Jacobs et al., 1991) where the mean and covariance matrix are generated by deep neural networks.

We propose Expected Information Maximization (EIM) [1], a novel approach capable of computing the I-projection between the model and the data. By exploiting the structure of the I-projection, we can derive a variational upper bound objective, which was previously used in the context of variational inference (Maaløe et al., 2016; Ranganath et al., 2016; Arenz et al., 2018). In order to work with this upper bound objective based on samples, we use a discriminator to approximate the required density ratio, relating our approach to GANs (Goodfellow et al., 2014; Nowozin et al., 2016; Uehara et al., 2016). The discriminator also allows us to use additional discriminative features to improve model quality. In our experiments, we demonstrate that EIM is much more effective in computing the I-projection than recent GAN approaches. We apply EIM to a synthetic obstacle avoidance task, an inverse kinematic task of a redundant robot arm as well as a pedestrian and car prediction task using the Stanford Drone Dataset (Robicquet et al., 2016) and a traffic dataset from the Next Generation Simulation program.

## 2 PRELIMINARIES

Our approach heavily builds on minimizing Kullback-Leibler divergences as well as the estimation of density ratios. We will therefore briefly review both concepts.

**Density Ratio Estimation.** Our approach relies on estimating density ratios $r(x) = q(x)/p(x)$ based on samples of $q(x)$ and $p(x)$. Sugiyama et al. (2012) introduced a framework to estimate such density ratios based on the minimization of Bregman divergences (Bregman, 1967). For our work we employ one approach from this framework, namely density ratio estimation by binary logistic regression. Assume a logistic regressor $C(x) = \sigma(\phi(x))$ with logits $\phi(x)$ and sigmoid activation function $\sigma$. Further, we train $C(x)$ on predicting the probability that a given sample $x$ was sampled from $q(x)$. It can be shown that such a logistic regressor using a cross-entropy loss is optimal for $C(x) = q(x)/(q(x) + p(x))$. Using this relation, we can compute the log density ratio estimator by

$$\log \frac{q(x)}{p(x)} = \log \frac{q(x)/(q(x) + p(x))}{p(x)/(q(x) + p(x))} = \log \frac{C(x)}{1 - C(x)} = \sigma^{-1}(C(x)) = \phi(x).$$

The logistic regressor is trained by minimizing the binary cross-entropy

$$\text{argmin}_{\phi(x)} \text{BCE}(\phi(x), p(x), q(x)) = -\mathbb{E}_{q(x)}\left[\log\left(\sigma(\phi(x))\right)\right] - \mathbb{E}_{p(x)}\left[\log\left(1 - \sigma(\phi(x))\right)\right],$$

where different regularization techniques such as $\ell_2$ regularization or dropout (Srivastava et al., 2014) can be used to avoid overfitting.

---

[1]Code available at `https://github.com/pbecker93/ExpectedInformationMaximization`

**Moment and Information Projection.** The Kullback-Leibler divergence (Kullback & Leibler, 1951) is a standard similarity measure for distributions. It is defined as $\text{KL}\left(p(x)||q(x)\right) = \int p(x)\log p(x)/q(x)dx$. Due to its asymmetry, the Kullback-Leibler Divergence provides two different optimization problems (Bishop, 2006) to fit a model distribution $q(x)$ to a target distribution $p(x)$, namely

$$\underbrace{\text{argmin}_{q(x)}\text{KL}\left(p(x)||q(x)\right)}_{\text{Moment-projection}} \quad \text{and} \quad \underbrace{\text{argmin}_{q(x)}\text{KL}\left(q(x)||p(x)\right)}_{\text{Information-projection}}.$$

Here, we will assume that $p(x)$ is the data distribution, i.e., $p(x)$ is unknown but we have access to samples from $p(x)$. It can easily be seen that computing the M-projection to the data distribution is equivalent to maximizing the likelihood (ML) of the model (Bishop, 2006). ML solutions match the moments of the model with the moments of the target distribution, which results in averaging over modes that can not be represented by the model. In contrast, the I-projection forces the learned generator $q(x)$ to have low probability whenever $p(x)$ has low probability, which is also called zero forcing.

## 3 RELATED WORK

We will now discuss competing methods for computing the I-projection that are based on GANs. Those are, to the best of our knowledge, the only other approaches capable of computing the I-projection solely based on samples of the target distribution. Furthermore, we will distinguish our approach from approaches based on variational inference that also use the I-projection.

**Variational Inference.** The I-projection is a common objective in Variational Inference (Opper & Saad, 2001; Bishop, 2006; Kingma & Welling, 2013). Those methods aim to fit tractable approximations to intractable distributions of which the unnormalized density is available. EIM, on the other hand, does not assume access to the unnormalized density of the target distributions but only to samples. Hence, it is not a variational inference approach, but a density estimation approach. However, our approach uses an upper bound that has been previously applied to variational inference (Maaløe et al., 2016; Ranganath et al., 2016; Arenz et al., 2018). EIM is especially related to the VIPS algorithm (Arenz et al., 2018), which we extend from the variational inference case to the density estimation case. Additionally, we introduce conditional latent variable models into the approach.

**Generative Adversarial Networks.** While the original GAN approach minimizes the Jensen-Shannon Divergence (Goodfellow et al., 2014), GANs have since been adapted to a variety of other distance measures between distributions, such as the Wasserstein distance (Arjovsky et al., 2017), symmetric KL (Chen et al., 2018) and arbitrary $f$-divergences (Ali & Silvey, 1966; Nowozin et al., 2016; Uehara et al., 2016; Poole et al., 2016). Since the I-projection is a special case of an $f$-divergence, those approaches are of particular relevance to our work. Nowozin et al. (2016) use a variational bound for $f$-divergences (Nguyen et al., 2010) to derive their approach, the $f$-GAN. Uehara et al. (2016) use a bound that directly follows from the density ratio estimation under Bregman divergences framework introduced by Sugiyama et al. (2012) to obtain their $b$-GAN. While the $b$-GAN's discriminator directly estimates the density ratio, the $f$-GAN's discriminator estimates an invertible mapping of the density ratio. Yet, in the case of the I-projection, both the $f$-GAN and the $b$-GAN yield the same objective, as we show in Appendix C.2. For both the $f$-GAN and $b$-GAN the desired $f$-divergence determines the discriminator objective. Uehara et al. (2016) note that the discriminator objective, implied by the I-projection, is unstable. As both approaches are formulated in a general way to minimize any $f$-divergence, they do not exploit the special structure of the I-projection. Exploiting this structure permits us to apply a tight upper bound of the I-projection for latent variable models, which results in a higher quality of the estimated models.

Li et al. (2019) introduce an adversarial approach to compute the I-projection based on density ratios, estimated by logistic regression. Yet, their approach assumes access to the unnormalized target density, i.e., they are working in a variational inference setting. The most important difference to GANs is that we do not base EIM on an adversarial formulation and no adversarial game has to be solved. This removes a major source of instability in the training process, which we discuss in more detail in Section 4.3.

## 4 EXPECTED INFORMATION MAXIMIZATION

Expected Information Maximization (EIM) is a general algorithm for minimizing the I-projection for any latent variable model. We first derive EIM for general marginal latent variable models, i.e., $q(x) = \int q(x|z)q(z)dz$ and subsequently extend our derivations to conditional latent variable models, i.e., $q(x|y) = \int q(x|z, y)q(z|y)dz$. EIM uses an upper bound for the objective of the marginal distribution. Similar to Expectation-Maximization (EM), our algorithm iterates between an M-step and an E-step. In the corresponding M-step, we minimize the upper bound and in the E-step we tighten it using a variational distribution.

### 4.1 EIM FOR LATENT VARIABLE MODELS

The I-projection can be simplified using a (tight) variational upper bound (Arenz et al., 2018) which can be obtained by introducing an auxiliary distribution $\tilde{q}(z|x)$ and using Bayes rule

$$\mathrm{KL}\left(q(x)||p(x)\right) = \underbrace{U_{\tilde{q},p}(q)}_{\text{upper bound}} - \underbrace{\mathbb{E}_{q(x)}[\mathrm{KL}\left(q(z|x)||\tilde{q}(z|x)\right)]}_{\geq 0},$$

where

$$U_{\tilde{q},p}(q) = \iint q(x|z)q(z)\left(\log\frac{q(x|z)q(z)}{p(x)} - \log\tilde{q}(z|x)\right)dzdx. \quad (1)$$

The derivation of the bound is given in Appendix B. It is easy to see that $U_{\tilde{q},p}(q)$ is an upper bound as the expected KL term is always non-negative. In the corresponding E-step, the model from the previous iteration, which we denote as $q_t(x)$, is used to tighten the bound by setting $\tilde{q}(z|x) = q_t(x|z)q_t(z)/q_t(x)$. In the M-step, we update the model distribution by minimizing the upper bound $U_{\tilde{q},p}(q)$. Yet, opposed to Arenz et al. (2018), we cannot work directly with the upper bound since it still depends on $\log p(x)$, which we cannot evaluate. However, we can reformulate the upper bound by setting the given relation for $\tilde{q}(z|x)$ of the E-step into Eq. 1,

$$U_{q_t,p}(q) = \int q(z)\left(\int q(x|z)\log\frac{q_t(x)}{p(x)}dx + \mathrm{KL}\left(q(x|z)||q_t(x|z)\right)\right)dz + \mathrm{KL}\left(q(z)||q_t(z)\right). \quad (2)$$

The upper bound now contains a density ratio between the old model distribution and the data. This density ratio can be estimated using samples of $q_t$ and $p$, for example, by using logistic regression as shown in Section 2. We can use the logits $\phi(x)$ of such a logistic regressor to estimate the log density ratio $\log(q_t(x)/p(x))$ in Equation 2. This yields an upper bound $U_{q_t,\phi}(q)$ that depends on $\phi(x)$ instead of $p(x)$. Optimizing this bound corresponds to the M-step of our approach. In the E-step, we set $q_t$ to the newly obtained $q$ and retrain the density ratio estimator $\phi(x)$. Both steps formally result in the following bilevel optimization problem

$$q_{t+1} \in \mathrm{argmin}_{q(x)}U_{q_t,\phi^*}(q) \quad \text{s.t.} \quad \phi^*(x) \in \mathrm{argmin}_{\phi(x)}\mathrm{BCE}(\phi(x), p(x), q_t(x)).$$

Using a discriminator also comes with the advantage that we can use additional discriminative features $g(x)$ as input to our discriminator that are not directly available for the generator. For example, if $x$ models trajectories of pedestrians, $g(x)$ could indicate whether the trajectory reaches any positions that are not plausible such as rooftops or trees. These features simplify the discrimination task and can therefore improve our model accuracy which is not possible with M-projection based algorithms such as EM.

### 4.2 EIM FOR CONDITIONAL LATENT VARIABLE MODELS

For conditional distributions, we aim at finding the conditional I-projection

$$\mathrm{argmin}_{q(x|y)}\mathbb{E}_{p(y)}\left[\mathrm{KL}\left(q(x|y)||p(x|y)\right)\right].$$

The derivations for the conditional upper bound follow the same steps as the derivations in the marginal case, where all distributions are extended by the context variable $y$. We refer to the supplement for details. The log density ratio estimator $\phi(x, y)$ now discriminates between samples of the joint distribution of $x$ and $y$. For training $\phi(x, y)$ we generate a new sample $x$ for each context $y$, using the distribution $q_{\text{old}}(x|y)$. Hence, as the context distribution is the same for the true data and the generated data, the log density ratio of the conditional distributions is equal to the log density ratio of the joint distributions.

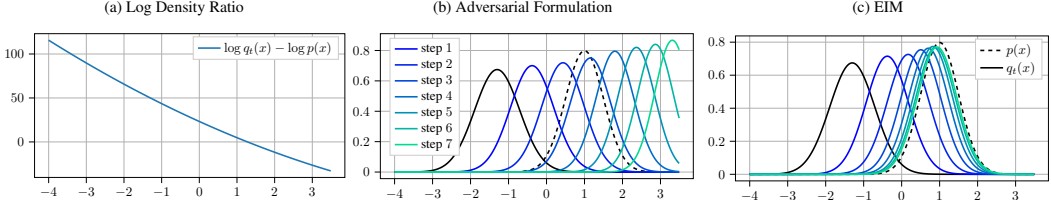

Figure 2: An illustrative example of the benefits of EIM versus an adversarial formulation. (a): The true log density ratio of the model and the target distribution (both are Gaussians). The location of the optimum is unbounded. (b): In the adversarial formulation, the generator minimizes the expected log density ratio. If we neglect that the adversarial discriminator changes with every update step of the generator, the generator updates yield an unbounded solution. Hence, too aggressive updates of the generator yield unstable behavior. (c): The upper bound of EIM introduces an additional KL-term as objective. Optimizing this objective directly yields the optimal solution without the need to recompute the density ratio estimate.

### 4.3 RELATION TO GANS AND EM

There is a close relation of EIM to GANs due to the use of a logistic discriminator for the density ratio estimation. It is therefore informative to investigate the differences in the case without latent variables. In an adversarial formulation, the density ratio estimator would directly replace the density ratio in the original I-projection equation, i.e.,

$$\text{argmin}_{q(x)} \int q(x)\phi^*(x)dx \quad \text{s.t.} \quad \phi^*(x) \in \text{argmin}_{\phi(x)}\text{BCE}(\phi(x), p(x), q^*(x)).$$

However, such adversarial games are often hard to optimize. In contrast, EIM offers a bilevel optimization problem where the discriminator is explicitly learned on the old data distribution $q_t(x)$,

$$\text{argmin}_{q(x)} \int q(x)\phi^*(x)dx + \text{KL}\left(q(x)||q_t(x)\right) \text{ s.t. } \phi^*(x) \in \text{argmin}_{\phi(x)}\text{BCE}(\phi(x), p(x), q_t(x)).$$

Thus, there is no circular dependency between the optimal generator and the optimal discriminator. Figure 2 illustrates that the proposed non-adversarial formulation does not suffer from too large model updates. Choosing the number and step-size of the updates is thus far less critical.

EIM can also be seen as the counter-part of Expectation-Maximization (EM). While EM optimizes the M-projection with latent variable models, EIM uses the I-projection. However, both approaches decompose the corresponding projections into an upper bound (or lower bound for EM) and a KL-term that depends on the conditional distribution $q(z|x)$ to tighten this bound. The exact relationship is discussed in Appendix C.1.

### 4.4 EIM FOR GAUSSIAN MIXTURES MODELS

We consider Gaussian mixture models with $d$ components, i.e., multivariate Gaussian distributions $q(\mathbf{x}|z_i) = \mathcal{N}(\boldsymbol{\mu}_i, \boldsymbol{\Sigma}_i)$ and a categorical distribution $q(z) = \text{Cat}(\boldsymbol{\pi})$ for the coefficients. As the latent distribution $q(z)$ is discrete, the upper bound in EIM (Equation 2) simplifies, as the integral over $z$ can be written as a sum. Similar to the EM-algorithm, this objective can be updated individually for the coefficients and the components. For both updates, we will use similar update rules as defined in the VIPS algorithm (Arenz et al., 2018). VIPS uses a trust region optimization for the components and the coefficients, where both updates can be solved in closed form as the components are Gaussian. The trust regions prevent the new model from going too far away from $q_t$ where the density ratio estimator is inaccurate, and hence, further stabilize the learning process. We will now sketch both updates, where we refer to Appendix B.2 for the full details.

For updating the coefficients, we assume that the components have not yet been updated, and therefore KL $\left(q(\mathbf{x}|z_i)||q_t(\mathbf{x}|z_i)\right) = 0$ for all $z_i$. The objective for the coefficients thus simplifies to

$$\text{argmin}_{q(z)} \sum_{i=1}^{d} q(z_i)\phi(z_i) + \text{KL}\left(q(z)||q_t(z)\right) \quad \text{with} \quad \phi(z_i) = \mathbb{E}_{q(\mathbf{x}|z_i)}\left[\phi(\mathbf{x})\right], \quad (3)$$

where $\phi(z_i)$ can be approximated using samples from the corresponding component. This objective can easily be optimized in closed form, as shown in the VIPS algorithm (Arenz et al., 2018). We also use a KL trust-region to specify the step size of the update. For updating the individual components, the objective simplifies to

$$\text{argmin}_{q(\mathbf{x}|z_i)}\mathbb{E}_{q(\mathbf{x}|z_i)}\left[\phi(\mathbf{x})\right] + \text{KL}\left(q(\mathbf{x}|z_i)||q_t(\mathbf{x}|z_i)\right). \tag{4}$$

As in VIPS, this optimization problem can be solved in closed form using the MORE algorithm (Abdolmaleki et al., 2015). The MORE algorithm uses a quadratic surrogate function that locally approximates $\phi(\mathbf{x})$. The resulting solution optimizes Equation 4 under a KL trust-region. The pseudo-code of EIM for GMMs can be found in Appendix A.

### 4.5 EIM FOR GAUSSIAN MIXTURES OF EXPERTS

In the conditional case, we consider mixtures of experts consisting of $d$ multivariate Gaussians, whose parameters depend on an input $\mathbf{y}$ in a nonlinear fashion, i.e., $q(\mathbf{x}|z_i, \mathbf{y}) = \mathcal{N}(\psi_{\mu,i}(\mathbf{y}), \psi_{\Sigma,i}(\mathbf{y}))$ and the gating is given by a neural network with softmax output. We again decompose the resulting upper bound into individual update steps for the components and the gating. Yet, closed-form solutions are no longer available and we need to resort to gradient-based updates. The objective for updating the gating is given by

$$\text{argmin}_{q(z|\mathbf{y})}\sum_{i=1}^{d}\left(\mathbb{E}_{p(\mathbf{y})q(\mathbf{x}|z_i,\mathbf{y})}\left[q(z_i|\mathbf{y})\phi(\mathbf{x},\mathbf{y})\right]\right) + \mathbb{E}_{p(\mathbf{y})}\left[\text{KL}\left(q(z|\mathbf{y})||q_t(z|\mathbf{y})\right)\right]. \tag{5}$$

We minimize this equation w.r.t. the parameters of the gating by gradient descent using the Adam (Kingma & Ba, 2014) algorithm. The objective for updating a single component $i$ is given by

$$\text{argmin}_{q(\mathbf{x}|z_i,\mathbf{y})}\mathbb{E}_{\tilde{p}(\mathbf{y}|z_i)}\left[\mathbb{E}_{q(\mathbf{x}|z_i,\mathbf{y})}\left[\phi(\mathbf{x},\mathbf{y})\right] + \text{KL}\left(q(\mathbf{x}|z_i,\mathbf{y})||q_t(\mathbf{x}|z_i,\mathbf{y})\right)\right], \tag{6}$$

where $\tilde{p}(\mathbf{y}|z_i) = p(\mathbf{y})q(z_i|\mathbf{y})/q(z_i)$. Note that we normalized the objective by $q(z_i) = \int p(\mathbf{y})q(z_i|\mathbf{y})d\mathbf{y}$ to ensure that also components with a low prior $q(z_i)$ get large enough gradients for the updates. As we have access to the derivatives of the density ratio estimator w.r.t. $\mathbf{x}$, we can optimize Equation 6 with gradient descent using the reparametrization trick (Kingma & Welling, 2013) and Adam.

## 5 EVALUATION

We compare our approach to GANs and perform an ablation study on a toy task, with data sampled from known mixture models. We further apply our approach to two synthetic datasets, learning the joint configurations of a planar robot as well as a non-linear obstacle avoidance task, and two real datasets, namely the Stanford Drone Dataset (Robicquet et al., 2016) and a traffic dataset from the Next Generation Simulation program. A full overview of all hyperparameters and network architectures can be found in Appendix E.

### 5.1 COMPARISON TO GENERATIVE ADVERSARIAL APPROACHES AND ABLATION STUDY

We compare to the $f$-GAN which is the only other method capable of minimizing the I-projection solely based on samples. We use data sampled from randomly generated GMMs with different numbers of components and dimensionalities. To study the influence of the previously mentioned differences of EIM to generative adversarial approaches, we also perform an ablation study. We compare to a version of EIM where we neglect the additional KL-term (EIM, no KL), a version were we trained all components and the coefficients jointly using gradient descent (Joint EIM), and a version where we do both (Joint EIM, no KL). The average I-projection achieved by the various approaches can be found in Figure 3.

### 5.2 LINE REACHING WITH PLANAR ROBOT

We extended the introductory example of the planar reaching task and collected expert data from a 10-link planar robot tasked with reaching a point on a line. We fitted GMMs with an increasing

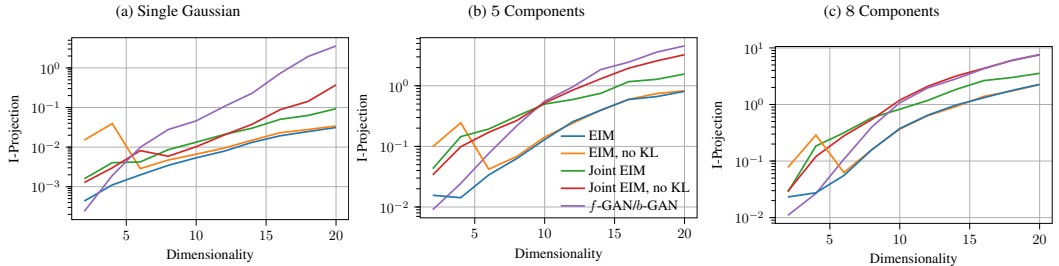

Figure 3: Average I-projection achieved for EIM, the $f$-GAN, and the modified EIM versions. The task is to fit a model to samples from a randomly generated GMM of different dimensions. Both the model and the target GMM have the same number of components. EIM clearly outperforms the generative adversarial approaches, especially for larger dimensions. The ablation study shows that the separated, closed-form updates clearly yield better results. Neglecting the KL has a big influence for lower dimensions, but is out-weighted by the error of the discriminator at higher dimensions.

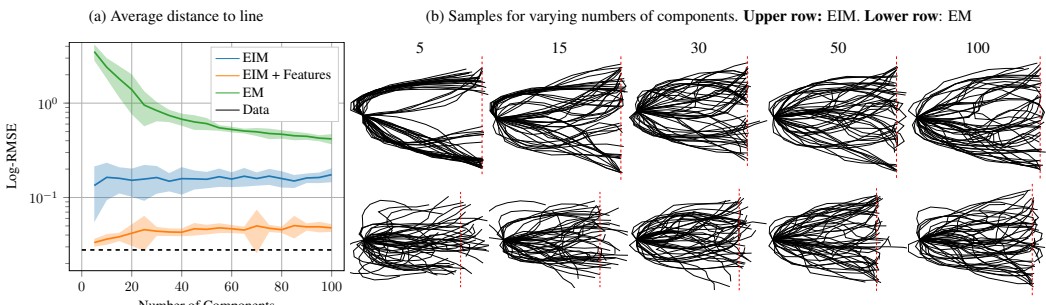

Figure 4: Average distance to line and samples for robot line reaching. While EIM for small numbers of components ignores modes, not considering the whole line, it learns models that achieve the underlying task, i.e., reach the line. Providing additional information to the density ratio estimator further decreases the average distance to the line. EM, on the other hand, averages over the modes, and thus, fails to reach the line even for large numbers of components.

number of components using EIM, EIM with additional features, where the end-effector coordinates for a given joint configuration were provided, and EM. Even for a large number of components, we see effects similar to the introductory example, i.e., the M-projection solution provided by EM fails to reach the line while EIM manages to do so. For small numbers, EIM ignores parts of the line, while more and more parts of it get covered as we increase the number of components. With the additional features, the imitation of the line reaching was even more accurate. See Figure 4, for the average distance between the end-effector and the line as well as samples from both EM and EIM.

## 5.3 PEDESTRIAN AND TRAFFIC PREDICTION

We evaluated our approach on data from the Stanford Drone Dataset (SDD) (Robicquet et al., 2016) and a traffic dataset from the Next Generation Simulation (NGS) program[2]. The SDD data consists of trajectories of pedestrians, bikes, and cars and we only used the data corresponding to a single video of a single scene (Video 1, deathCircle). The NGS data consists of trajectories of cars where we considered the data recorded on Lankershim Boulevard. In both cases we extracted trajectories of length 5, yielding highly multimodal data due to pedestrians, bikes, and cars moving at different speeds and in different directions. We evaluated on the achieved log-likelihood of EIM and EM, see Figure 5. EM achieves the highest likelihood as it directly optimizes this measure. However, we can already see that EM massively overfits when we increase the number of components as the test-set likelihood degrades. EIM, on the other hand, produced better models with an increasing number of components. Additionally, we generated a mask indicating whether a given point is on the road or

---

[2] https://data.transportation.gov/Automobiles/Next-Generation-Simulation-NGSIM-Vehicle-Trajector/8ect-6jqj

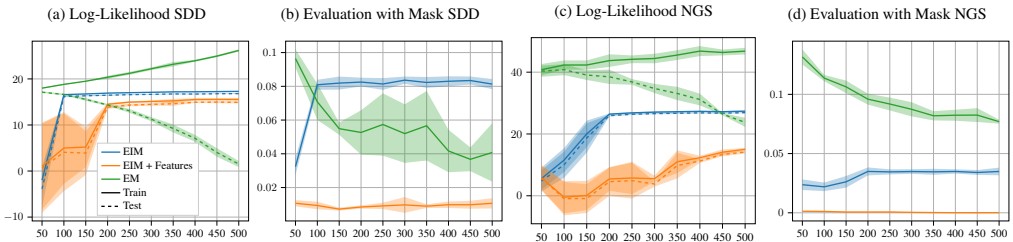

Figure 5: Results on the traffic prediction tasks. Naturally, EM achieves the highest training log-likelihood. Yet, for large numbers of components, a severe amount of overfitting is observed. EIM, on the other hand, has no problems working with high numbers of components and achieves a higher test log-likelihood, despite optimizing a different objective. We also provided a 'road mask' as additional features for the discriminator. We used this road mask to evaluate how realistic the generated samples are. While EIM without features produced more realistic samples on the Lankershim dataset, we needed the feature input to outperform EM on this evaluation on the SDD dataset.

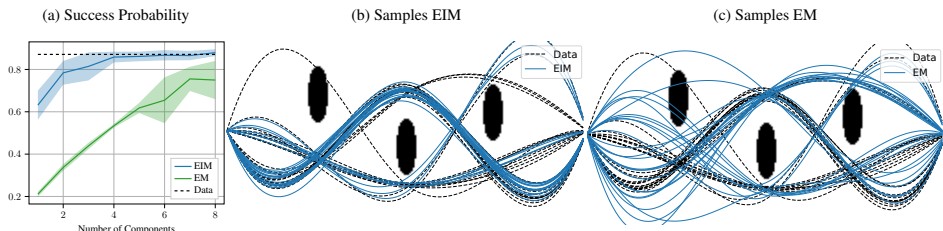

Figure 6: Results on the obstacle avoidance task. (a): Even for a small number of components, EIM has a rather high probability of success, i.e., placing a trajectory that does not hit any obstacle. Even with a sufficient number of components, i.e., eight, EM fails to achieve good results. (b) and (c): Samples of a mixture with 4 components, learned by EIM and EM respectively. EM clearly averages over multiple modes in the data distribution.

not and evaluated how realistic the learned models are by measuring the amount of samples violating the mask, i.e., predicting road users outside of the road. We also evaluate a version of EIM where we provide additional features indicating if the mask is violated. EIM achieves a much better value on this mask for the NGS dataset, while we needed the additional mask features for the discriminator on the SDD dataset to outperform EM. Both experiments show that EIM can learn highly multi-modal density estimates that produce more realistic samples than EM. They further show that the models learned by EIM can be refined by additional prior knowledge provided as feature vectors.

## 5.4 OBSTACLE AVOIDANCE

We evaluate the conditional version of EIM on an artificial obstacle avoidance task. The context contains the location of three obstacles within an image. The gating, as well as the components, are given by deep neural networks. Details about the network architectures can be found in the Appendix. The data consists of trajectories going from the left to the right of the image. The trajectories are defined by setting 3 via-points such that no obstacle is hit. To generate the data we sample via-points over and under the obstacles with a probability proportional to the distance between the obstacle and the image border. Hence, for three obstacles, there are $2^3 = 8$ different modes in the data. Note that, like in most real-world scenarios, the expert data is not perfect, and about $13\%$ of the trajectories in the dataset hit an obstacle. We fit models with various numbers of components to this data using EIM and EM and compare their performance on generating trajectories that achieve the goal. Results are shown in Figure 6 together with a visualization of the task and samples produced by EIM and EM. EIM was able to identify most modes for the different given inputs and did not suffer from any averaging effect. In contrast, EM does not find all modes. As a consequence, some

components of the mixture model had to average over multiple modes, resulting in poor quality trajectories.

# 6    Conclusion

We introduced Expected Information Maximization (EIM), a novel approach for computing the I-projection between general latent variable models and a target distribution, solely based on samples of the latter. General upper bound objectives for marginal and conditional distributions were derived, resulting in an algorithm similar to EM, but tailored for the I-projection instead of the M-projection. We introduced efficient methods to optimize these upper bound objectives for mixture models. In our experiments, we demonstrated the benefits of the I-projection for different behavior modelling tasks. The introduced approach opens various pathways for future research. While we focused on mixture models, the derived upper bounds are not exclusive to those and can be used for arbitrary latent variable models. Another possibility is an online adaptation of the number of used components. Arenz et al. (2018) propose heuristics for such an adaptation in their VIPS approach. Those could easily be adapted to our approach.

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

# A    PSEUDO CODE

---

EIM-for-GMMs($\{\mathbf{x}_p^{(j)}\}_{j=1\cdots N}, q(\mathbf{x})$);

**Input:** Data $\{\mathbf{x}_p^{(j)}\}_{j=1\cdots N}$, Initial Model $q(\mathbf{x}) = \sum_{i=1}^d q(\mathbf{x}|z_i)q(z_i) = \sum_{i=1}^d \pi_i \mathcal{N}(\mathbf{x}|\boldsymbol{\mu}_i, \boldsymbol{\Sigma}_i)$

**for** $i$ *in number of iterations* **do**

    **E-Step:**

    $q_t(z) = q(z)$, $q_t(\mathbf{x}|z_i) = q(\mathbf{x}|z_i)$ for all components $i$

    **Update Density Ratio Estimator:**

    sample data from model $\{\mathbf{x}_q^{(j)}\}_{j=1\cdots N} \sim q_t(\mathbf{x})$

    retrain density ratio estimator $\phi(\mathbf{x})$ on $\{\mathbf{x}_p^{(j)}\}_{j=1\cdots N}$ and $\{\mathbf{x}_q^{(j)}\}_{j=1\cdots N}$

    **M-Step Coefficients:**

    **for** $i$ *in number of components* **do**

        compute loss $l_i = \frac{1}{N} \sum_{j=1}^N \phi\left(\mathbf{x}_q^{(j)}\right)$ with samples $\{\mathbf{x}_q^{(j)}\}_{j=1\cdots N} \sim q_t(\mathbf{x}|z_i)$

    **end**

    update $q(z)$ using losses $l_i$ and MORE equations

    **M-Step Components:**

    **for** $i$ *in number of components* **do**

        fit $\hat{\phi}(\mathbf{x})$ surrogate to pairs $\left(\mathbf{x}_q^{(j)}, \phi\left(\mathbf{x}_q^{(j)}\right)\right)$ with samples $\{\mathbf{x}_q^{(j)}\}_{j=1\cdots N} \sim q_t(\mathbf{x}|z_i)$

        update $q(\mathbf{x}|z_i)$ using surrogate $\hat{\phi}(\mathbf{x})$ and MORE equations

    **end**

**end**

**Algorithm 1:** Expected Information Maximization for Gaussian Mixture Models.

---

Pseudo-code for EIM for GMMs can be found in algorithm 1

# B    DERIVATIONS

Derivations of the upper bound stated in Equation 1. We assume latent variable models $q(x) = \int q(x|z)q(z)dz$ and use the identities $q(x|z)q(z) = q(z|x)q(x)$ and $\log q(x) = \log q(x|z)q(z) - \log q(z|x)$.

$$
\begin{aligned}
\text{KL}\left(q(x)||p(x)\right) &= \int q(x) \log \frac{q(x)}{p(x)}dx = \iint q(x|z)q(z) \log \frac{q(x)}{p(x)}dzdx \\
&= \iint q(x|z)q(z)\left(\log \frac{q(x|z)q(z)}{p(x)} - \log q(z|x)\right)dzdx \\
&= \iint q(x|z)q(z)\left(\log \frac{q(x|z)q(z)}{p(x)} - \log q(z|x) + \log \tilde{q}(z|x) - \log \tilde{q}(z|x)\right)dzdx \\
&= \iint q(x|z)q(z)\left(\log \frac{q(x|z)q(z)}{p(x)} - \log \tilde{q}(z|x)\right)dzdx - \iint q(x|z)q(z)\left(\log q(z|x) - \log \tilde{q}(z|x)\right)dzdx \\
&= \iint q(x|z)q(z)\left(\log \frac{q(x|z)q(z)}{p(x)} - \log \tilde{q}(z|x)\right)dzdx - \int q(x) \int q(z|x) \log \frac{q(z|x)}{\tilde{q}(z|x)}dzdx \\
&= U(q, \tilde{q}, p) - \mathbb{E}_{q(x)}\left[\text{KL}\left(q(z|x)||\tilde{q}(z|x)\right)\right].
\end{aligned}
$$

After plugging the E-Step, i.e., $\tilde{q}(z|x) = q_t(x|z)q_t(z)/q_t(x)$, into the objective it simplifies to

$$
\begin{aligned}
& U(q, \tilde{q}, p) \\
&= \iint q(x|z)q(z) \left( \log \frac{q(x|z)q(z)}{p(x)} - \log \frac{q_t(x|z)q_t(z)}{q_t(x)} \right) dzdx \\
&= \iint q(x|z)q(z) \left( \log q(x|z) + \log q(z) - \log p(x) - \log q_t(x|z) - \log q_t(z) + \log q_t(x) \right) dzdx \\
&= \iint q(x|z)q(z) \left( \log \frac{q_t(x)}{p(x)} + \log \frac{q(x|z)}{q_t(x|z)} + \log \frac{q(z)}{q_t(z)} \right) dzdx \\
&= \int q(z) \left( \int q(x|z) \left( \log \frac{q_t(x)}{p(x)} + \log \frac{q(x|z)}{q_t(x|z)} \right) dx + \log \frac{q(z)}{q_t(z)} \right) dz \\
&= \int q(z) \int q(x|z) \log \frac{q_t(x)}{p(x)} dxdz + \int q(z) \int q(x|z) \log \frac{q(x|z)}{q_t(x|z)} dxdz + \int q(z) \log \frac{q(z)}{q_t(z)} dz \\
&= \iint q(x|z)q(z) \log \frac{q_t(x)}{p(x)} dzdx + \mathbb{E}_{q(z)} \left[ \mathrm{KL}\left( q(x|z) || q_t(x|z) \right) \right] + \mathrm{KL}\left( q(z) || q_t(z) \right),
\end{aligned}
$$

which concludes the derivation of upper bound of latent variable models.

## B.1 DERIVATIONS CONDITIONAL UPPER BOUND

By introducing an auxiliary distribution $\tilde{q}(z|x, y)$ the upper bound to the expected KL for conditional latent variable models $q(x|y) = \int q(x|z, y)q(z|y)dz$ can be derived by

$$
\begin{aligned}
\mathbb{E}_{p(y)} \left[ \mathrm{KL}\left( q(x|y) || p(x|y) \right) \right] &= \iint p(y)q(x|y) \log \frac{q(x|y)}{p(x|y)} dxdy \\
&= \int p(y) \iint q(x|z, y)q(z|y) \left( \log \frac{q(x|z, y)q(z|y)}{p(x|y)} - \log q(z|x, y) \right) dzdxdy \\
&= \int p(y) \iint q(x|z, y)q(z|y) \\
&\quad \cdot \left( \log \frac{q(x|z, y)q(z|y)}{p(x|y)} - \log q(z|x, y) + \log \tilde{q}(z|x, y) - \log \tilde{q}(z|x, y) \right) dzdxdy \\
&= \int p(y) \iint q(x|z, y)q(z|y) \left( \log \frac{q(x|z, y)q(z|y)}{p(x|y)} - \log \tilde{q}(z|x, y) \right) dzdxdy \\
&\quad - \int p(y) \iint q(x|z, y)q(z, y) \left( \log q(z|x, y) - \log \tilde{q}(z|x, y) \right) dzdxdy \\
&= \int p(y) \iint q(x|z, y)q(z|y) \left( \log \frac{q(x|z, y)q(z|y)}{p(x|y)} - \log \tilde{q}(z|x, y) \right) dzdxdy \\
&\quad - \iint p(y)q(x|y) \int q(z|x, y) \log \frac{q(z|x, y)}{\tilde{q}(z|x, y)} dzdxdy \\
&= U(q, \tilde{q}, p) - \mathbb{E}_{p(y), q(x|y)} \left[ \mathrm{KL}\left( q(z|x, y) || \tilde{q}(z|x, y) \right) \right].
\end{aligned}
$$

During the E-step the bound is tightened by setting $\tilde{q}(z|x,y) = q_t(x|z,y)q_t(z|y)/q_t(x|y)$.

$$U(q, \tilde{q}, p)$$

$$= \int p(y) \iint q(x|z,y)q(z|y) \left( \log \frac{q(x|z,y)q(z|y)}{p(x|y)} - \log \frac{q_t(x|z,y)q_t(z|y)}{q_t(x|y)} \right) dzdxdy$$

$$= \int p(y) \iint q(x|z,y)q(z|y)$$
$$\cdot \left( \log q(x|z,y) + \log q(z|y) - \log p(x|y) - \log q_t(x|z,y) - \log q_t(z|y) + \log q_t(x|y) \right) dzdxdy$$

$$= \int p(y) \iint q(x|z,y)q(z|y) \left( \log \frac{q_t(x|y)}{p(x|y)} + \log \frac{q(x|z,y)}{q_t(x|z,y)} + \log \frac{q(z|y)}{q_t(z|y)} \right) dzdxdy$$

$$= \int p(y) \int q(z|y) \left( \int q(x|z,y) \left( \log \frac{q_t(x|y)}{p(x|y)} + \log \frac{q(x|z,y)}{q_t(x|z,y)} \right) dx + \log \frac{q(z|y)}{q_t(z|y)} \right) dzdy$$

$$= \int p(y) \int q(z|y) \int q(x|z,y) \log \frac{q_t(x|y)}{p(x|y)} dxdzdy$$
$$+ \int p(y) \int q(z|y) \int q(x|z,y) \log \frac{q(x|z,y)}{q_t(x|z,y)} dxdzdy + \int p(y) \int q(z|y) \log \frac{q(z|y)}{q_t(z|y)} dzdy$$

$$= \iiint p(y)q(z|y)q(x|z,y) \log \frac{q_t(x|y)}{p(x|y)} dxdzdy$$
$$+ \mathbb{E}_{p(y),q(z|y)} \left[ \text{KL} \left( q(x|z,y) || q_t(x|z,y) \right) \right] + \mathbb{E}_{p(y)} \left[ \text{KL} \left( q(z|y) || q_t(z|y) \right) \right],$$

which concludes the derivation of the upper bound for conditional latent variable models.

## B.2 USING MORE FOR CLOSED FORM UPDATES FOR GMMS

The MORE algorithm, as introduced by Abdolmaleki et al. (2015), can be used to solve optimization problems of the following form

$$\text{argmax}_{q(x)} \mathbb{E}_{q(x)}[f(x)] \quad \text{s.t.} \quad \text{KL} \left( q(x) || q_{\text{old}}(x) \right) \leq \epsilon$$

for an exponential family distribution $q(x)$, some function $f(x)$, and an upper bound on the allowed change, $\epsilon$. Abdolmaleki et al. (2015) show that the optimal solution is given by

$$q(x) \propto q_{\text{old}}(x) \exp \left( \frac{f(x)}{\eta} \right) = \exp \left( \frac{\eta \log q_{\text{old}}(x) + f(x)}{\eta} \right),$$

where $\eta$ denotes the Lagrangian multiplier corresponding to the KL constraint. In order to obtain this Lagrangian multiplier, the following, convex, dual function has to be minimized

$$g(\eta) = \eta \epsilon + \eta \log \int \exp \left( \frac{\eta \log q_{\text{old}}(x) + f(x)}{\eta} \right) dx. \tag{7}$$

For discrete distributions, such as the categorical distribution used to represent the coefficients of a GMM, we can directly work with those equations. For continuous distributions, Abdolmaleki et al. (2015) propose approximating $f(x)$ with a local surrogate. The features to fit this surrogate are chosen such that they are compatible (Kakade, 2002), i.e., of the same form as the distributions sufficient statistics. For multivariate Gaussians, the sufficient statistics are squared features and thus the surrogate compatible to such a Gaussian distribution is given by

$$\hat{f}(x) = -\frac{1}{2} \mathbf{x}^T \hat{\mathbf{F}} \mathbf{x} + \hat{\mathbf{f}}^T \mathbf{x} + f_0.$$

The parameters of this surrogate can now be used to update the natural parameters of the Gaussian, i.e, the precision matrix $\mathbf{Q} = \mathbf{\Sigma}^{-1}$ and $\mathbf{q} = \mathbf{\Sigma}^{-1} \boldsymbol{\mu}$ by

$$\mathbf{Q} = \mathbf{Q}_t + \frac{1}{\eta} \hat{\mathbf{F}} \quad \text{and} \quad \mathbf{q} = \mathbf{q}_t + \frac{1}{\eta} \hat{\mathbf{f}}.$$

In order to apply the MORE algorithm to solve the optimization problems stated in Equation 3 and Equation 4 we make two trivial modifications. First, we invert the signs in Equation 3 and Equation 4, as we are now maximizing. Second, to account for the additional KL term in our objectives, we add 1 to $\eta$, everywhere except the first term of the sum in Equation 7.

## C  ELABORATION ON RELATED WORK

### C.1  RELATION BETWEEN EIM AND EM

Recall that the Expectation-Maximization (EM) algorithm (Dempster et al., 1977) maximizes the log-likelihood of the data by iteratively maximizing and tightening the following lower bound

$$\mathbb{E}_{p(x)}\left[\log q(x)\right] = \mathbb{E}_{p(x)}\left[\int \tilde{q}(z|x)\log\frac{q(x,z)}{\tilde{q}(z|x)}dz\right] + \mathbb{E}_{p(x)}\left[\int \tilde{q}(z|x)\log\frac{\tilde{q}(z|x)}{q(z|x)}dz\right]$$

$$= \underbrace{\mathcal{L}(q,\tilde{q})}_{\text{lower bound}} + \underbrace{\mathbb{E}_{p(x)}\left[\text{KL}\left(\tilde{q}(z|x)||q(z|x)\right)\right]}_{\geq 0}.$$

It is instructive to compare our upper bound (Equation 1) to this lower bound. As mentioned, maximizing the likelihood is equivalent to minimizing the M-projection, i.e., $\text{argmin}_{q(x)}\text{KL}\left(p(x)||q(x)\right)$, where, in relation to our objective, the model and true distribution have switched places in the non-symmetric KL objective. Like our approach, EM introduces an auxiliary distribution $\tilde{q}(z|x)$ and bounds the objective from below by subtracting the KL between auxiliary distribution and model, i.e., $\text{KL}\left(\tilde{q}(z|x)||q(z|x)\right)$. In contrast, we obtain our upper bound by adding $\text{KL}\left(q(z|x)||\tilde{q}(z|x)\right)$ to the objective. Again, the distributions have exchanged places within the KL.

### C.2  EQUALITY OF $f$-GAN AND $b$-GAN

As pointed out in section 3 both the $f$-GAN (Nowozin et al., 2016) and the $b$-GAN (Uehara et al., 2016) yield the same objective for the I-projection.

We start with the $f$-GAN. Nowozin et al. (2016) propose the following adversarial objective, based on a variatonal bound for $f$-divergences (Nguyen et al., 2010)

$$\text{argmin}_{q(x)}\text{argmax}_{V(x)}F(q(x),V(x)) = \mathbb{E}_{p(x)}\left[g(V(x))\right] - \mathbb{E}_{q(x)}\left[f^*(g(V(x)))\right].$$

Here $V(x)$ denotes a neural network with linear output, $g(v)$ the output activation and $f^*(t)$ the Fenchel conjugate (Hiriart-Urruty & Lemaréchal, 2012) of $f(u)$, i.e., the generator function of the $f$-divergence. For the I-projection $f(u) = -\log u$ and $f^*(t) = -1 - \log(t)$. In theory, the only restriction posed on the choice of $g(v)$ is that it outputs only values within the domain of $f^*(t)$ Nowozin et al. (2016) suggest $g$'s for various $f-$ divergences and chose exclusively monotony increasing functions which output large values for samples that are believed to be from the data distribution. For the I-projection they suggest $g(v) = -exp(-v)$. Thus the $f$-GAN objective for the I-projection is given by

$$\text{argmin}_{q(x)}\text{argmax}_{V(x)}F(q(x),V(x)) = -\mathbb{E}_{p(x)}\left[\exp(-V(x)\right] + \mathbb{E}_{q(x)}\left[1 - V(x)\right].$$

The $b$-GAN objective follows from the density ratio estimation framework given by Sugiyama et al. (2012) and is given by

$$\text{argmin}_{q(x)}\text{argmax}_{r(x)}\mathbb{E}_{p(x)}\left[f'(r(x))\right] - \mathbb{E}_{q(x)}\left[f'(r(x))r(x) - f(r(x))\right]$$

Here $f'(u)$ denotes the derivative of $f(u)$ and $r(x)$ denotes an density ratio estimator. We need to enforce $r(x) > 0$ for all $x$ to obtain a valid density ratio estimate. In practice this is usually done by learning $r_l(x) = \log r(x)$ instead. Plugging $r_l(x)$, $f(u)$ and $f'(u) = 1/u$ into the general $b$-GAN objective yields

$$\text{argmin}_{q(x)}\text{argmax}_{r_l(x)}F(q(x),r_l(x)) = -\mathbb{E}_{p(x)}\left[\exp(-r_l(x))\right] + \mathbb{E}_{q(x)}\left[1 - r_l(x)\right].$$

Which is the same objective as the $f$-GAN uses. Yet, $f$-GAN and $b$-GAN objectives are not identical for arbitrary $f$-divergences.

## D  VISUALIZATION OF SAMPLES

### D.1  PEDESTRIAN AND TRAFFIC PREDICTION

Samples from the Stanford Drone Dataset can be found in Figure 7

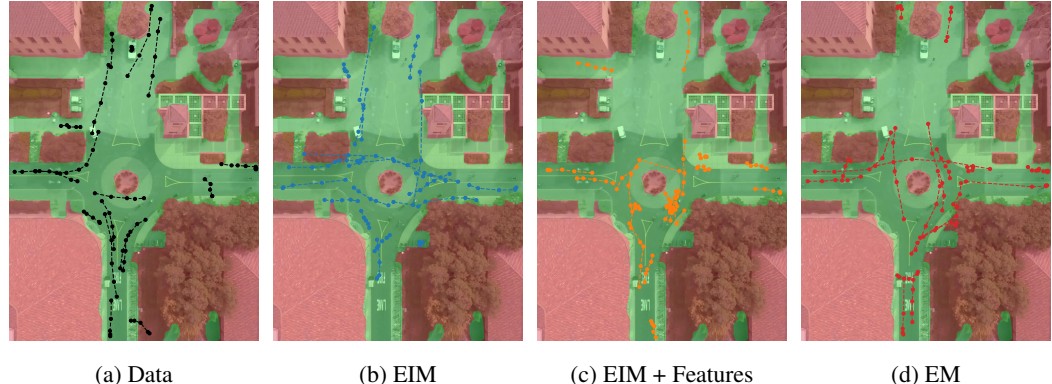

(a) Data          (b) EIM          (c) EIM + Features          (d) EM

Figure 7: Samples from the Dataset, EIM, EIM with features and EM, plotted over the reference image from the Stanford Drone Dataset and the generated mask. In the mask green corresponds to valid regions and red to invalid regions. EIM with the additional feature input generates samples that stay within the 'road mask' and are therefore considered to be more realistic.

## E    HYPERPARAMETERS

In all experiments, we realize the density ratio estimator as fully connected neural networks which we train using Adam (Kingma & Ba, 2014) and early stopping using a validation set.

**Comparison to Generative Adversarial Approaches and Ablation Study**

- Data: $10,000$ Train Samples, $5,000$ Test Samples, $5,000$ Validation samples (for early stopping the density ratio estimator)
- Density Ratio Estimator (EIM) / Variational function $V(x)$ ($f$-GAN): 3 fully connected layers, 50 neurons each, trained with L2 regularization with factor 0.001, early stopping and batch size $1,000$
- Updates EIM: MORE-like updates with $\epsilon = 0.05$ for components and coefficients, $1,000$ samples per component and update
- Updates FGAN: Iterate single update steps for generator and discriminator using learning rates of $1e-3$ and batch size of $1,000$.

**Line Reaching with Planar Robot**

- Data: $10,000$ train samples, $5,000$ test samples, $5,000$ validation samples (for early stopping the density ratio estimator)
- Density Ratio Estimation: 2 fully connected layers of width 100, early stopping and batch size $1,000$
- Updates: MORE-like updates with $\epsilon = 0.005$ for components and coefficients, $1,000$ samples per component and update

**Pedestrian and Traffic Prediction**

- Data SDD: $7,500$ train samples, $3,801$ test samples, $3,801$ validation samples (for early stopping the density ratio estimator)
- Data NGS: $10,000$ train samples, $5,000$ test samples, $5,000$ validation samples (for early stopping the density ratio estimator)
- Density Ratio Estimation: 3 fully connected layers of width 256, trained with L2 regularization with factor 0.0005 early stopping and batch size $1,000$.
- Updates: MORE-like updates with $\epsilon = 0.01$ for components and coefficients, $1,000$ samples per component and update

**Obstacle Avoidance**

- Data: $1,000$ train contexts with $10$ samples each, $500$ test contexts with $10$ samples each, $500$ validation contexts with $10$ samples each (for early stopping the density ratio estimator).

- Density Ratio Estimation: 3 fully connected layers of width $256$, trained with L2 regularization with factor $0.0005$ early stopping and batch size $1,000$.

- Component and Gating networks: 2 fully connected layer of width $64$ for each component and the gating. Trained with Adam ($\alpha = 1e - 3, \beta_0 = 0.5$) for $10$ epochs in each iteration.

