# OpenReview forum: "Expected Information Maximization: Using the I-Projection for Mixture Density Estimation"
_ICLR.cc/2020/Conference — Accept (Poster)_

### Official Review · AnonReviewer2 · 2019-10-23
**Official Blind Review #2**

**Rating:** 6

**Review:**

The paper presents an algorithm to match two distributions with latent variables, named expected information maximization (EIM). Specifically, EIM is based on the I-Projection, which basically is equivalent to minimizing the reverse KL divergence (i.e. min KL[p_model || p_data]); to handle latent variables, an upper-bound is derived, which is the corresponding reverse KL divergence in the joint space. To minimize that joint reverse KL, a specific procedure is developed, leading to the presented EIM. EIM variants for different applications are discussed. Fancy robot-related experiments are used to evaluate the presented algorithm.

Overall, the paper is in good shape wrt the logic and the writing. My main concerns focus on the novelty (compared to existing methods that are similar but not discussed) and the experiments. For the former, reverse KL has been exploited before, both in the marginal space [1] and the joint one [2]. Other detailed comments are listed below.

As Eq 4 is for matching two joint distributions, discussions/comparisons should be made to reveal the novelty of the presented EIM over existing methods such as [2], etc.

In Figure 2 (b), the experimental settings for adversarial learning are not fair, as the discriminator is not fixed there.

In Sec 4.4, it seems EIM is highly overlapped with VIPS. So what're the advantages of EIM here?

In Figure 3, how many steps for Generator and Discriminator are used for f-GAN? Does f-GAN finally converge? It would be helpful if some results are given to demonstrate the final state of each method.

In Eq. 9, adding the denominator q(z_i) will change the optimal solution. Why only add it to the first term?

In Section 5.3 and Figure 5, “SSD” might be a typo.

[1] Adversarial Learning of a Sampler Based on an Unnormalized Distribution. AISTATS 2018.
[2] Symmetric Variational Autoencoder and Connections to Adversarial Learning. AISTATS 2019.


**Experience Assessment:**

I have published one or two papers in this area.

**Review Assessment: Checking Correctness Of Derivations And Theory:**

I carefully checked the derivations and theory.

**Review Assessment: Checking Correctness Of Experiments:**

I assessed the sensibility of the experiments.

**Review Assessment: Thoroughness In Paper Reading:**

I read the paper thoroughly.

---

> ### Author Response · Authors · 2019-11-08
> **Response to Reviewer 2**
>
> We thank the reviewers for their time and valuable feedback. Besides fixing small typos, ambiguities and unclearities we elaborated on the relation and differences to previously existing GAN and VI methods (see in particular section 2).
>
>  Also, we uploaded an implementation of EIM, which can be found at: https://github.com/eimAuthors/EIM
>
> We are now going to answer your specific questions:
>
> “My main concerns focus on the novelty” / “For the former, reverse KL has been exploited before, both in the marginal space [1] and the joint one [2]. Other detailed comments are listed below.” / “In Sec 4.4, it seems EIM is highly overlapped with VIPS. So what're the advantages of EIM here?” -
>
> Our approach is, to the best of our knowledge, the first approach allowing non-adversarial computation of the I-Projection based solely on samples of the target distribution.
>
> The difference to VIPS and [1]  is that both assume access to the unnormalized (log) density of the target distribution, i.e. are applicable for variational inference. EIM on the other hand assumes access to samples of the target distribution, i.e. is applicable for density estimation.
>
> The difference to [1] and [2] is that EIM is not adversarial as pointed out in section 4.3.
>
> We reworked the related work section to make these important distinctions clearer.
>
> Furthermore, [2] introduces a bound for the symmetric KL between the joints over x and z (where x is the random variable underlying the target samples and z the latent variable). The learned discriminator thus needs both x and z as inputs. In order to infer the latent variable for the target samples an additional variational distribution q(z|x) (i.e. an encoder) needs to be learned.
> EIM does not use the symmetric KL, but the reverse KL. Furthermore it works with a bound for the KL between the marginals over x, not the joint of x and z. Thus the discriminator only needs to be given x and the latent variable z does not need to be inferred for the training data, i.e., no “encoder” is necessary.
>
> “In Figure 2 (b), the experimental settings for adversarial learning are not fair, as the discriminator is not fixed there. “
>
> The purpose of this figure (and section 4.3 in general) is to provide an illustrative example of the immediate effects and benefits of avoiding the adversarial forumulation
>
> “In Figure 3, how many steps for Generator and Discriminator are used for f-GAN? Does f-GAN finally converge?[...]” -
>
> Figure 3: As suggested in the f-GAN paper we alternate single generator and discriminator steps. We also evaluated training the discriminator longer without notable changes to the final performance. The f-GANs do eventually converge and we report the best value achieved on a test set for each run, averaged over 20 runs. A list of all hyperparameters can be found in the appendix, we added the parameters for the f-GAN training to this.
>
> “In Eq. 9, adding the denominator q(z_i) will change the optimal solution. Why only add it to the first term?” -  The notation in this equation was a bit unclear. The denominator is in fact added to both terms, which scales the optimal value but does not change the optimal solution. We apologize for the unclear notation and adapted the equation to make it clearer.
>
> “In Section 5.3 and Figure 5, “SSD” might be a typo. “ - Indeed a typo, thanks for pointing it out, we fixed it.
>
> We hope we could clarify and remove some of the remaining doubts in our approach. We invite you to ask additional questions and engage in further discussion if this is not the case.

---

### Official Review · AnonReviewer3 · 2019-10-26
**Official Blind Review #3**

**Rating:** 6

**Review:**

This paper propose EIM an analog to EM but to perform the I-projection (i.e. reverse-KL) instead of the usual M-projection for EM. The motivation is that the reverse-KL is mode-seeking in contrast to the forward-KL which is mode-covering. The authors argue that in the case that the model is mis-specified, I-projection is sometimes desired as to avoid putting mass on very unlikely regions of the space under the target p.

The authors propose an iterative procedure that alternates between estimating likelihood ratios and proposal distribution by minimizing an upper bound on the reverse-KL. The derivations seem correct. There are some experiments, majoritarily in the robotics domain. As the author point out, likelihood shouldn't be the right metric since you are now minimizing the reverse-KL---I would have liked the authors to spend some more time on the right way to evaluate---and actually use that new metric. Finally, there has been plethora of work on different objectives and distance between distributions as well as a zoo of lower/upper bounds on how to evaluate them---it would be interesting to have more connections to prior work.

[Pros]
- clearly written
- clear motivation
- correct derivations
- interesting algorithm

[Cons]
- experiments are a little weak (and focus on a single domain)
- would have liked to see an explicit algorithm for the optimization procedure
- small lack of clarity in the presentation of Section 4.1---notation q_t is not introduced for example
- more discussion about the evaluation metric
- linking it more to prior work


**Experience Assessment:**

I have published one or two papers in this area.

**Review Assessment: Checking Correctness Of Derivations And Theory:**

I carefully checked the derivations and theory.

**Review Assessment: Checking Correctness Of Experiments:**

I assessed the sensibility of the experiments.

**Review Assessment: Thoroughness In Paper Reading:**

I read the paper at least twice and used my best judgement in assessing the paper.

---

> ### Author Response · Authors · 2019-11-08
> **Response to Reviewer 3**
>
> We thank the reviewers for their time and valuable feedback. Besides fixing small typos, ambiguities and unclearities we elaborated on the relation and differences to previously existing GAN and VI methods (see in particular section 2).
>
>  Also, we uploaded an implementation of EIM, which can be found at: https://github.com/eimAuthors/EIM
>
> We are now going to answer your specific questions:
>
> “I would have liked the authors to spend some more time on the right way to evaluate”/ “more discussion about the evaluation metric”
>
> We believe that the reverse KL is the right metric for the applications that we consider, which motivates the formulation of our optimization problem.
> Sadly it is not possible to compute the reverse KL if the true density underlying the data is not known (which is the case in all but the first experiment, in which we evaluated the reverse KL). This is also the key reason for why minimizing the reverse KL is much harder than minimizing the forward KL (i.e. maximizing the likelihood).
>
> Thus, we had to resort to auxillary metrics. The likelihood is an obvious choice here since it is a standard evaluation criterion for generative models and easy to compute. Additionally we evaluate our models on metrics, meaningful to the task at hand. By combining the likelihood with the auxiliary metric, we can evaluate both whether the data distribution is covered and whether unrealistic samples are generated by the learned model. The same is typically done for GAN approaches which suffer from the same problem of a non-computable objective for non-toy tasks.
>
> “- would have liked to see an explicit algorithm for the optimization procedure” - There is pseudocode for the GMM case in the appendix.  As previously mentioned, we also released the real code by now.
>
> “- small lack of clarity in the presentation of Section 4.1---notation q_t is not introduced for example” - We kindly ask you to elaborate on this lack of clarity so we might clarify. We already clarified the introduction of q_t.
>
> “- linking it more to prior work” - the related work section has been reworked.
>
>
> We hope we could clarify and remove some of the remaining doubts in our approach. We invite you to ask additional questions and engage in further discussion if this is not the case.

---

> > ### Comment · AnonReviewer3 · 2019-11-15
> > **--**
> >
> > I have read the authors' answers and appreciate the time spent writing the rebuttal. I will maintain my initial assessment.

---

### Official Review · AnonReviewer1 · 2019-11-03
**Official Blind Review #1**

**Rating:** 6

**Review:**

In this paper, the authors proposed a new algorithm -- expected information maximization (EIM) -- for computing the I-projection of the data distribution to the model distribution, solely based on samples for general latent variable models, where the paper only focus on Gaussian mixtures models and experts. The proposed method applies a variational upper bound to the I-projection objective which is decomposable for each mixture components and the coefficients. Overall, I think the proposed technique quite sound and results are convincing. However, I do have some questions:

Questions:
-- The proposed EIM algorithm in Sec 4.1 seems to require “re-training” the discriminator every time the q function is updated. How this can be applicable to more realistic and complex models where training requires millions of gradient steps? Will the same algorithm can be applied on more general latent variable models or even implicit models like GAN does? As the paper has pointed out, the vanilla f-GAN itself can be seen as optimizing some forms of the I-Projection (reverse Kullback-Leibler divergence) objective.
-- I am a little confused about Sec 4.3. It seems that the latent variable z is not necessary for the proposed EIM?
-- Also, the typical practise of training GAN is also iterative between the generator and the discriminator, we sometimes need to update the discriminator with more steps than the generator? Shouldn’t it be the exactly same as the proposed EIM except we have an additional regularization term of KL(q(x) || q_t(x)) which might be the true reason why training gets more stable than standard GAN?
-- Similar to the previous two questions, how to compute the regularization term KL(q(x) || q_t(x)) in EIM for normal generator which is typically implicit?


**Experience Assessment:**

I do not know much about this area.

**Review Assessment: Checking Correctness Of Derivations And Theory:**

I assessed the sensibility of the derivations and theory.

**Review Assessment: Checking Correctness Of Experiments:**

I assessed the sensibility of the experiments.

**Review Assessment: Thoroughness In Paper Reading:**

I read the paper at least twice and used my best judgement in assessing the paper.

---

> ### Author Response · Authors · 2019-11-08
> **Response to Reviewer 1**
>
> We thank the reviewers for their time and valuable feedback. Besides fixing small typos, ambiguities and unclearities we elaborated on the relation and differences to previously existing GAN and VI methods (see in particular section 2).
>
>  Also, we uploaded an implementation of EIM, which can be found at https://github.com/eimAuthors/EIM
>
> We are now going to answer your specific questions:
>
> “How this can be applicable to more realistic and complex models where training requires millions of gradient steps?” - The density ratio estimator does not need to be trained from scratch every iteration. It can be warm started using the density ratio estimator from the previous iteration. We use early-stopping for regularization which will typically end training after a few iterations. The change in density ratio during each iteration is rather small since the model updates are constraint.
>
>   “I am a little confused about Sec 4.3. It seems that the latent variable z is not necessary for the proposed EIM?” - In Sec 4.3 we consider only the simplest possible model, a single univariate Gaussian (i.e. a Gaussian Mixture with one component) for illustrative purposes. Like for any latent variable approach the latent variable can be “omitted” by choosing q(z) to be a deterministic distribution.
>
> “Also, the typical practise of training GAN [...] training gets more stable than standard GAN?” - We in fact update the discriminator for multiple steps with early stopping. The difference to GANs is that our approach is not adversarial, this removes a key reason for the instability of GAN training. Our derivations for this non-adversarial optimization are based on the additional KL term. Hence, we also believe that it is a major reason for the improved stability of EIM.
>
> “Will the same algorithm can be applied on more general latent variable models” -
> The general approach derived in section 4 can be applied to general latent variable models.
> (For discussion on the KL term see the next bullet point)
>
> “ or even implicit models like GAN does?” / “[...] , how to compute the regularization term KL(q(x) || q_t(x)) in EIM for normal generator which is typically implicit?” -
> If the KL term can not be computed in closed form it can still be approximated using samples as long as the model density is tractable. Note that we do not need to compute the KL divergence between the marginals KL(q(x) || q_t(x)) but only the KL between the conditionals KL(q(x|z) || q_t(x|z)) and the latent distribution KL(q(z)||q_t(z)). Even if the Marginal q(x) is intractable (as for GANs) the density of the conditionals is usually not.
> Nethertheless, investigating how to use EIM for typical GAN scenarios, e.g., generating image data, is an interesting direction for future work.
>
> We hope we could clarify and remove some of the remaining doubts in our approach. We invite you to ask additional questions and engage in further discussion if this is not the case.

---

### Decision · Program_Chairs · 2019-12-19

**Decision:**

Accept (Poster)

**Comment:**

The paper proposes a new algorithm called Expected Information Maximization (EIM) for learning latent variable models while computing the I-projection solely based on samples. The reviewers had several questions, which the authors sufficiently answered. The reviewers agree that the paper should be accepted. The authors should carefully read the reviewer questions and comments and use them to improve their final manuscript.